# An Intuitionistic Fuzzy-Rough Set-Based Classification for Anomaly Detection

**Fokrul Alom Mazarbhuiya** [1,*] and **Mohamed Shenify** [2,*]

1   School of Fundamental and Applied Sciences, Assam Don Bosco University, Guwahati 782402, India
2   College of Computer Science and IT, Albaha University, Al Baha 65799, Saudi Arabia
*   Correspondence: fokrul.mazarbhuiya@dbuniversity.ac.in (F.A.M.); maalshenify@bu.edu.sa (M.S.)

**Abstract:** The challenging issues of computer networks and databases are not only the intrusion detection but also the reduction of false positives and increase of detection rate. In any intrusion detection system, anomaly detection mainly focuses on modeling the normal behavior of the users and detecting the deviations from normal behavior, which are assumed to be potential intrusions or threats. Several techniques have already been successfully tried for this purpose. However, the normal and suspicious behaviors are hard to predict as there is no precise boundary differentiating one from another. Here, rough set theory and fuzzy set theory come into the picture. In this article, a hybrid approach consisting of rough set theory and intuitionistic fuzzy set theory is proposed for the detection of anomaly. The proposed approach is a classification approach which takes the advantages of both rough set and intuitionistic fuzzy set to deal with inherent uncertainty, vagueness, and indiscernibility in the dataset. The algorithm classifies the data instances in such a way that they can be expressed using natural language. A data instance can possibly or certainly belong to a class with degrees of membership and non-membership. The empirical study with a real-world and a synthetic dataset demonstrates that the proposed algorithm has normal true positive rates of 91.989% and 96.99% and attack true positive rates of 91.289% and 96.29%, respectively.

**Keywords:** intuitionistic fuzzy sets; fuzzy correlation; fuzzy relation; α-cut of a fuzzy relation; similarity relation; fuzzy lower and upper approximation of sets





## 1. Introduction

Anomaly detection (AD) can be termed as the detection of the patterns that deviate from the expected normal behavior [1]. Anomaly detection is essential when such abnormality in the datasets can provide sufficient system information [2]. An anomaly may be malicious activities, instrumentation errors, human errors, etc. It is an emerging research area with applications in fields such as cybersecurity, medicine, intrusion detections, financial fraud, etc. With the advancement of computers and networks and their extensive uses, organizations are becoming vulnerable to malicious activities. Although the existing defense mechanism can provide protection up to a reasonable extent, the malicious attackers are becoming more sophisticated in intruding across the networks. In the case of internal attack, it might be interesting and challenging to identify the anomalies. The detection of anomaly from network data has been accepted as one of the most promising research areas of information security.

Intrusion detection systems (IDSs) [3] are security devices for shielding networks or systems from unauthorized activity that could endanger accessibility, privacy, or integrity. Anomaly-detection-based and signature-recognition-based techniques are the two main categories of IDSs. By monitoring the systems and categorizing the actions as normal or anomalous, the former [4] is utilized to find network and computer misuse or intrusions. Anomaly-based IDS is the name given to the ensuing system [3,4]. However, a signature-recognition-based intrusion detection technique [5] uses a database of known

attack signatures and raises an alarm whenever network traffic matches any signature. Usually, a computer and associated network can easily use an anomaly-based IDS as a risk mitigation technique.

Several anomaly detection approaches were developed in the previous decade [6–11]. The classification-based approach is one such approach. The classification [12] is a data processing tool to classify the items into predefined classes, and it has been applied in several areas such as pattern recognition, anomaly detection, prediction, machine vision, etc. [13–16]. An anomaly detection algorithm using neighborhood rough set classification for dataset with mixed attributes was proposed in [13]. In [14], the authors proposed a decision-tree-based approach for the detection of anomaly in the results of computer assessment to improve the quality of educational management. In [15], the authors proposed a Bayesian-network-based anomaly detection method. In [16], the authors developed a single deep RBF network, used to predict control actions and to detect hostile cyber physical system attacks. In [17], the authors presented an anomaly detection method using a rough-set-based attribute reduction. In [18], the authors introduced an anomalous event identification approach on video surveillance applications. In [19], the authors introduced a neural-network-based semi-supervised approach for efficient anomaly detection.

A problem similar to classification approach is also addressed using the clustering approach [20–24]. In [25], the authors proposed a complex method for detecting anomaly from real-time data using recurrence and fractal analysis. In [26], the authors made a detailed comparative analysis of five different time series models of anomaly detection. In [27], an ensemble learning model was applied to investigate and forecast outliers of the enormous system logs. In [28], the authors suggested a strategy for anomaly detection that permits the use of state-of-the-art feature selection techniques for idea representation of meta-features. A new outline focusing on data-technology-based real-time AD was proposed in [29], which uses a streaming sliding window factor corset clustering algorithm. In [30], the authors proposed a mixed clustering algorithm (MCAD) for detecting anomaly in real-time data. In [31], the authors proposed an approach called density-increasing path (DIP) to address issues of arbitrary shapes and unknown cluster numbers appearing during clustering processes.

Most of the aforesaid methods only addressed the accuracy of the anomaly detection and a few addressed the false positive rates of the methods. Since the increase in the false positive rate decreases the detection rates and thus the efficacy of any classifier, it is required to minimize the false positive rates. Again, the normal and anomalous behaviors of the system are difficult to predict as there is no precise boundary differentiating one from another. In this scenario, either fuzzy set theory or rough set theory, or the combination of both, can effectively be utilized.

L. A. Zadeh [32] introduced fuzziness in the realm of mathematics by formally defining it as a generalization of ordinary set. Atanassov [33] introduced intuitionistic fuzzy sets (IFSs) by generalizing them in terms of membership and nonmembership functions. Most of the works on anomaly detection used Zadeh's [32] fuzzy set, and a few only used IFS. Since the IFS has the inherent ability to tackle the imbalance and overlapping data [34], it can efficiently be used to describe the uncertainty, imprecision, and vagueness in more generalized ways than the traditional fuzzy approaches [32]. Considering the strength of the intuitionistic fuzzy sets, in [35], the authors proposed an intuitionistic approach to detect anomaly from time series data. In [36], the authors proposed the formula for correlation coefficient of intuitionistic fuzzy sets whose value lies in the interval [0, 1]. Fuzzy relation, $\alpha$-cut of a fuzzy relation, and fuzzy equivalence relations were introduced in [37,38].

Pawlak [39] introduced the rough set theory to deal with uncertainty, imprecision, or vagueness that exist in the datasets. Using the features of an equivalence relation, [40] nicely applied the rough-set-based classification to discrete datasets. In [41], the authors proposed an efficient method using fuzzy neighborhood rough set for the detection anomaly in large datasets. In [42], the authors proposed an efficient fuzzy-rough-set-based algorithm for

feature selection. In [43,44], the authors proposed two density-based approaches using neutrosophic sets and fuzzy proximity relations for the detection anomaly. In [45], the authors proposed an NN classification algorithm which uses the fuzzy-rough lower and upper approximations to classify test objects, or to predict their decision value. The methods discussed above used the parameters such as entropy and weighted density as classification criterion for anomaly detection. However, when using correlation coefficient of intuitionistic fuzzy sets, classification rules can be generated where each data instance participating in the rules is characterized by its membership as well as non-membership values defined over a universe of discourse.

Thivagar et al. [46] introduced nano topological space with respect to a subset $X$ of universe $U$ in terms of lower and upper approximation of $X$. In [47], the authors not only introduced a nano topology structure but also applied it in medical diagnosis. In [48], the authors introduced three novel fuzzy nano topologies. Most classification-based anomaly detection algorithms developed up until today used different well-known measures to differentiate classes, and very few works were reported using the statistical measures such as correlation coefficient. Secondly, most of the fuzzy-rough approaches consider the corresponding fuzziness in Zadeh's sense [32]. However, if the approach can be extended to the intuitionistic fuzzy set, then the detected anomalies can provide more information about the system.

In this article, a hybrid approach consisting of intuitionistic fuzzy set (IFS) and rough set (RS) was used in the classification algorithm for the anomaly detection of network datasets. The objectives of the paper are threefold.

- First of all, a formula for correlation coefficient of IFSs is defined.
- Secondly, using the above correlation coefficient, an $\alpha$-relation (for a preassigned value of $\alpha$) and an equivalence relation [49–51] are generated to generate two approximations.
- Finally, a classification-based hybrid algorithm (IFRSCAD) consisting of both IFS and RS is proposed to generate the certain and possible fuzzy rules.

Furthermore, the proposed algorithm (IFRSCAD) is implemented using Matlab with two well-known datasets: KDDCUP'99 Network Anomaly Detection dataset [52] and Kinsune Network Attack dataset [53]. The classification results are compared with other classification-based methods, namely, Cuijuan et al. [17], Wang et al. [35], deep-RBF network [16], Bayes network [15], and decision tree [14]. It is found that the proposed algorithm is comparatively more efficient than others with respect to true positive rates and false positive rates. The time-complexity of the IFRSCAD is also compared with a well-known clustering-based algorithm MCAD [30] and is found to be comparatively efficient.

This paper is formatted in the following ways. The recent advances in this field are described in Section 2. The problem definition is given in Section 3. The algorithm and the flowchart explaining the system are given in Section 4. The time-complexity analysis is presented in Section 5. The experimental study and outcomes are presented in Section 6, and, lastly, the conclusions, limitations, and future directions of work are given in Section 7.

## 2. Related Works

AD [1] is termed as the discovery of those patterns that deviate from previously occurring ones. It can be useful for obtaining sufficient system information [2], and is one of the vital areas of modern research, which is receiving more and more attention of the researcher day by day. A couple of anomaly detection systems have already been developed [3,4]. Classification-based anomaly detection systems are some of the many. Using a classification-based labeling technique, Abdullah et al. [6] presented a method of anomaly detection in cellular networks. In [6], the authors used negative selection algorithm for detecting anomalies in multidimensional data. Taha et al. [8] reviewed the different anomaly detection methods for categorical data. Diaz Verdejo et al. [5] proposed an efficient alternative approach, named signature-recognition-based detection, in the context of web attacks.

Mazarbhuiya et al. [13] introduced a neighborhood rough-set-based classification approach to detect the anomaly in a mixed attribute dataset. For assessment of the computer and to improve the quality of educational management, a decision-tree-based anomaly detection was proposed [14]. A Bayesian-network-based algorithm for anomaly detection and offering correction hints was presented in [15]. In [16], the authors designed a single deep-RBF network to predict control actions and detect unwanted attacks in cyber physical systems. In [16], the authors proposed a rough set attribute reduction approach to detect anomaly. Wang et al. [17] designed an efficient intuitionistic fuzzy-set-based approach to detect anomaly from network traffic. Sengonul et al. [18] introduced AI-based analysis of anomaly detection in video surveillance applications. Fan et al. [19] introduced a neural-network-based semi-supervised approach for efficient anomaly detection.

Anomaly detection using a clustering approach was also studied by many researchers. Mazarbhuiya et al. [20] proposed an agglomerative hierarchical-clustering-based anomaly detection algorithm for anomaly detection in network datasets. An fuzzy $c$-means clustering-based anomaly detection method was proposed in [21]. Mazarbhuiya et al. [22] proposed a mixed algorithm consisting of features of both $k$-means and hierarchical algorithm for anomaly detection in network datasets. Retting et al. [23] proposed an algorithm of online anomaly detection in big data streams. Alguliyev et al. [24] proposed a clustering-based anomaly detection for big data. Using fractal and recurrence analysis, Alghawli et al. [25] proposed a real-time anomaly detection algorithm in time series data.

Kim et al. [26] performed a comparative analysis of five models of time series anomaly detection. In [27], the authors applied an ensemble learning model to study and predict anomaly of the enormous system logs. Halstead et al. [28] devised a strategy for anomaly detection that permitted the use of the latest feature selection techniques for idea representation of meta-features. Habeeb et al. [29] presented a data-technology-based framework focusing on real-time anomaly detection, which used a streaming sliding window factor corset clustering algorithm. Mazarbhuiya et al. [30] introduced a mixed clustering algorithm for anomaly detection of real-time data. Zhao et al. [31] proposed an efficient density-increasing path (DIP) anomaly detection approach to address arbitrary shapes and unknown cluster numbers appearing during clustering processes.

The fuzzy set was formally introduced by Zadeh [32] to deal with imprecision, uncertainty, or linguistic vagueness occurring in any dataset. Generalizing the concept of fuzzy set, Atanassov [33] defined intuitionistic fuzzy sets using membership and non-membership functions. Eulalia et al. [34] proposed an IFS-based classification on imbalance and overlapping classes to capture inherent imprecision, vagueness, and uncertainty occurring in the dataset. Wang et al. [35] proposed an intuitionistic fuzzy-set-based approach for the detection anomaly from time series data. Gerstenkorn et al. [36] proposed the definition correlation coefficient of intuitionistic fuzzy sets. Zadeh et al. [37] introduced the details of fuzzy similarity relations. In [38], the concepts of $\alpha$-cut of a fuzzy relation and fuzzy equivalence relations were introduced in detail.

Rough set theory was introduced by Pawlak [39] to deal with imprecision, uncertainty, or vagueness that exist in any datasets. Using properties to equivalence relation, Nowicki et al. [40] proposed a rough-set-based classification method on discrete datasets. Maroune et al. [41] proposed an anomaly-detection-based method on a highly scalable approach to compute the nearest neighbor of objects using rough set theory. Li et al. [42] proposed an efficient fuzzy-rough-set-based approach for the feature selection. Sangeetha et al. [43,44], proposed two density approaches based on neutrosophic sets and fuzzy proximity relations for the detection anomaly. Yuan et al. [45] introduced a neural-network-based classification algorithm using the fuzzy-rough lower and upper approximations to classify test objects or to predict their decision value.

Thivagar et al. [46,47] not only proposed the structure of nanotopological space in terms of lower and upper approximation but also applied it in medical diagnosis. Shumrani et al. [48] first introduced the concept of the covering-based rough fuzzy nanotopology, the covering-based rough intuitionistic fuzzy nanotopology, and the covering-

based rough neutrosophic nanotopology. In [49], the authors introduced the concept of fuzzy-rough set theory. Maji et al. [50] applied fuzzy-rough set for relevant genes selection from microarray data. Chimphlee et al. [51] proposed an anomaly-based IDS, which used fuzzy-rough clustering method. In [30], the authors conducted the experimental studies with two well-known datasets: KDDCUP'99 [52] Network Anomaly Detection dataset and Kitsune [53] Network Attack dataset.

## 3. Problem Definitions

Below, we present some important terms and definitions used in the paper.

**Definition 1.** *Fuzzy set.*

Let $X = \{x_1, x_2, \ldots x_n\}$ be the universe of discourse. A fuzzy set [32] $A$ on $X$ is characterized by

$$A = \{(x_i, \mu_A(x_i)); x_i \in X, i = 1, 2, \ldots n\} \tag{1}$$

where $\mu_A : X \to [0, 1]$, the membership function, gives the grade of membership of each element $x_i \in X$ in $A$.

**Definition 2.** *Intuitionistic fuzzy set.*

Atanassov [33] proposed the definition of an intuitionistic fuzzy set $A$ on $X$ as

$$A = \{(x_i, \mu_A(x_i), \nu_A(x_i)); x_i \in X, i = 1, 2, \ldots n\} \tag{2}$$

where $\mu_A : X \to [0, 1]$ and $\nu_A : X \to [0, 1]$ are the membership function and nonmembership function of the fuzzy set $A$, respectively, satisfying the condition $0 \leq \mu_A(x_i) + \nu_A(x_i) \leq 1$, for every $x_i \in X$.

**Definition 3.** *Correlation of intuitionistic fuzzy sets.*

Let $A = \{(x_i; \mu_A(x_i), \nu_A(x_i)); x_i \in X, i = 1, 2, \ldots n\}$ and $B = \{(x_i; \mu_B(x_i), \nu_B(x_i)); x_i \in X, i = 1, 2, \ldots n\}$ are two intuitionistic fuzzy sets on $X = \{x_1, x_2, \ldots x_n\}$. Gerstenkorn et al. [36] proposed the formula correlation coefficient as

$$\rho_{AB} = \frac{\sum_{i=1}^{n} [\mu_A(x_i).\mu_B(x_i) + \nu_A(x_i).\nu_B(x_i)]}{\sqrt{\sum_{i=1}^{n} [(\mu_A(x_i))^2 + (\nu_A(x_i))^2] \sum_{i=1}^{n} [(\mu_B(x_i))^2 + (\nu_B(x_i))^2]}} \tag{3}$$

Furthermore, $0 \leq \rho_{AB} \leq 1$.

**Definition 4.** *Fuzzy relation* [37,38].

For any data instances $x_i$; $i$ = 1, 2, $\ldots$ $m$ in $U$, we define a fuzzy relation $R$ on $U$ as $R = \{(x_i, x_j); \rho_{xixj}; x_i, x_j \in U\}$. Since $0 \leq \rho \leq 1$, $R$ will be an equivalence relation.

**Definition 5.** $\alpha - cut \, R_\alpha$ [37,38].

An $\alpha - $ cu t $R_\alpha$ of a fuzzy relation $R$ on $U$ is a crisp set containing the elements with membership values greater than $\alpha$ that is

$$R_\alpha = \{(x, y); \, \mu_R(x, y) \geq \alpha, \in (0, 1], \, x, y \in U\} \tag{4}$$

**Definition 6.** *α—relation* [37,38].

For any data instances $x_i$; $i = 1, 2, \ldots m$ in $U$ and $0 < \alpha \leq 1$, the $\alpha$—cut $R_\alpha$ of $R$ generates $\alpha$-relation $(U, \rho)$ as

$$\alpha(x_i) = \{x; \rho_{x_i x} \geq \alpha\}. \tag{5}$$

**Proposition 1.** [37,38]

If a fuzzy relation $R$ is an equivalence relation in max–min sense, then for $\alpha \in (0, 1)$, $R_\alpha$ possesses an equivalence relation. Therefore, any $\alpha$—relation represented by an $\alpha$—cut $R_\alpha$ will have an equivalence relation. The ordered pair $(U, R_\alpha)$ is an approximation space.

**Definition 7.** *Fuzzy-Rough Set.*

Fuzzy-rough set theory is an extension of rough set theory where the crisp equivalence class concept is extended to form fuzzy equivalence classes. Let the conditional and decision attributes of an information systems both be intuitionistic fuzzy sets and let us define an $\alpha$—relation in the aforesaid manner. Since a fuzzy equivalence relation generates a fuzzy partition of the universe of discourse, the $\alpha$—relation will generate a series of fuzzy equivalence classes [49–51], known as fuzzy knowledge granules. Letting $(U, R)$ represent a fuzzy approximation space and $X$ be a fuzzy subject of $U$ intuitionistic sense, the intuitionistic fuzzy nano lower approximation, the intuitionistic fuzzy nano upper approximation, and the intuitionistic nano boundary approximation of $X$ on $(U, R)$ are denoted by $\underline{I}(X)$, $\overline{I}(X)$, and $B_I(X)$, respectively, which are expressed as follows [48]:

$$\underline{I}(X) = \left\{ \left( x, \mu_{\underline{R}X}(x), \nu_{\underline{R}X}(x) \right), y \in [x]_R, x \in U \right\} \tag{6}$$

$$\overline{I}(X) = \left\{ \left( x, \mu_{\overline{R}X}(x), \nu_{\overline{R}X}(x) \right), y \in [x]_R, x \in U \right\} \tag{7}$$

$$B_I(X) = \overline{I}(X) - \underline{I}(X) \tag{8}$$

where $\mu_{\underline{R}X}(x) = inf_{y \in [x]_R}(y)$, $\nu_{\underline{R}X}(x) = sup_{y \in [x]_R}(y)$, $\mu_{\overline{R}X}(x) = sup_{y \in [x]_R}(y)$, and $\nu_{\overline{R}X}(x) = inf_{y \in [x]_R}(y)$.

## 4. Proposed Algorithm

For generating classification rules, we choose a suitable value of the correlation coefficient ($\alpha$) to define the $\alpha$-relation. The correlation coefficient used to define the relation is given in Section 3. The procedure of finding classification rules is given as follows. We have a collection of $m$-data instances, each of which is described by $n$-intuitionistic fuzzy attributes and is represented as an intuitionistic fuzzy matrix [54], where each entry is <$x_{ij}$, $y_{ij}$>, $x_{ij} \in [0, 1]$, $y_{ij} \in [0, 1]$, and $0 \leq x_{ij} + y_{ij} \leq 1$, $I = 1, 2, \ldots m$ and $j = 1, 2, \ldots n$. Usually, the dataset can be expressed as an information system $(U, C \cup D)$, where $C$ and $D$ are conditional and decision attributes, respectively, and are expressed as intuitionistic fuzzy sets. The method is described below.

The first step of the proposed method is to compute $\alpha$-relation of the conditional attribute using correlation coefficient, and compute the equivalence classes of decision attributes using the same formula of the correlation coefficient. The value of $\alpha$ is taken to be 0.4. Then, the "infimum" operator is applied on fuzzy knowledge granules of conditional attributes. Then, intuitionistic fuzzy nano lower approximation and intuitionistic fuzzy nano upper approximation are constructed using decision class. Then, the boundary regions are also found. With the help of two approximations, two sets of fuzzy rules, namely, the certain fuzzy rules and possible fuzzy rules, can be generated. The proposed method is also explained with the help of the flowchart given in Figure 1 below.

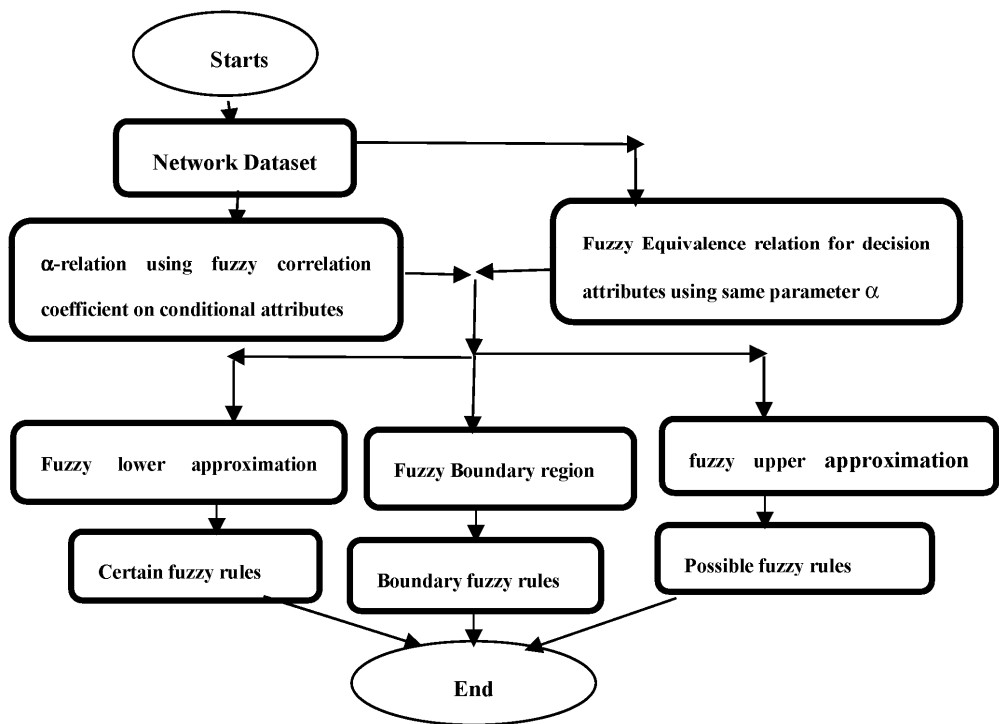

**Figure 1.** Flowchart of the algorithm.

The pseudocode for the algorithm is given as follows.

| **Algorithm 1:** IFRSCAD |
| --- |
| 1: Input ($U$, $C \cup D$), $\alpha / / C$, the conditional fuzzy attributes, $D$, the decision fuzzy attributes |
| 2: Step1. Create $\alpha$-relation on $C$ using correlation coefficient. |
| 3: Step2. Create the fuzzy equivalence relation for $D$. |
| 4: Step3. Apply 'infimum' operator on the fuzzy granules of records of $U$ brought up by $C$. |
| 5: Step4. Construct separately nano lower approximation space ($\underline{I}(X)$) Nano upper approximation space $\overline{I}(X)$ for $D$ and the result of fuzzy granules after applying 'infimum' to $C$. |
| 6: Step5. Find boundary regions. |
| 7: Step6. Generate certain fuzzy rules from nano lower approximation space, possible fuzzy rules from nano upper approximation, and boundary rules from boundary region. |

Obviously, each rule generated by the system is fuzzy in the intuitionistic sense. That is, attributes contributing in any of the rules will be in the intuitionistic fuzzy set.

## 5. Complexity Analysis

To generate $\alpha$-relation, the algorithm needs to choose all possible pairs of data instances from $U$, compute their correlation coefficients, then compare these with $\alpha$. These are performed in ($1/2 \ ^{|U|}C_2.|C| + 1/2 \ ^{|U|}C_2$), where ($1/2 \ ^{|U|}C_2$) computation is required for choosing pairs of data instances, $|C|$ is required for computation correlation coefficients, and ($1/2 \ ^{|U|}C_2$) number of comparisons with $\alpha$ are required. Thus, the running cost of step 1 is O($m^2.n$), where $|U| = m$ and $|C| = n$. For generating equivalence relation for $D$, the algorithm needs to take all possible pairs of data instances and compute the correlations, and this can be performed in ($1/2 \ ^{|U|}C_2.|C|$). The running cost for step 2 is O($m^2.n$). Thus, the running cost of step 1 and step 2 is O($m^2.n + m^2.n$) = O($m^2.n$). The running cost for step 3 is O($m$). For generating the nano topology, the lower approximation, upper approximation, and boundary regions of the set have to be generated, which takes the computational time of O($|X|.|U|$). Therefore, the total cost from step 1 to step 5 is O($m^2.n + m + |X|.|U|$) = O($m^2.n$), which is the worst-case complexity. Step 6 takes

constant time. Therefore, the overall time-complexity of the proposed algorithm is O($m^2.n$), which shows that the proposed algorithm is quite efficient.

## 6. Experimental Analysis and Results

### 6.1. Datasets

KDDCUP'99 [52]: It is a synthetic dataset that simulates intrusion in the military network environment. The data are collected for 9 weeks, and the training data consist of 5000 thousand network connections. The attributes can be divided into the classes, viz., normal (unauthorized access to local super user privileges, unauthorized access from a remote machine), dos, and probe.

Kitsune [53]: It is a group of nine network attack datasets, each containing millions of network packets and different cyberattacks, that were either gathered from an IP-based commercial surveillance system or a network of IoT devices.

The above datasets were acquired through the UCI machine repository. A brief description of the datasets is given in Table 1.

**Table 1.** Dataset descriptions.

| Dataset | Dataset Characteristics | Attribute Characteristics | No. of Instances | No. of Attributes |
|---------|------------------------|---------------------------|------------------|-------------------|
| KDDCUP'99 Network Anomaly Detection dataset [44] | Multivariate | Numeric, categorical, temporal | 4,898,431 | 41 |
| Kitsune Network Attack dataset [45] | Multivariate, sequential, time series | Real, temporal | 27,170,754 | 115 |

### 6.2. Experimental Results and Analysis

The experiments were carried out in Matlab with Intel Core i7-2600 machine with 3.4 GHz, 8 MB Cache, 8 GB RAM, and 500 GB hard disc, running Windows 10, and the outcomes were analyzed with five prominent classification-based methods, namely, Cuijuan et al.'s algorithm [17], Wang et al.'s algorithm [35], deep-RBF network [16], Bayes network [15], and decision tree [14]. The classifiers were built using the aforesaid dataset. The value of $\alpha$ was assumed to be 0.4. The classifiers were then used to categorize any new instance as either normal traffic or an attack. For a variety of attributes sizes, the outcomes of all the aforesaid six methods were recorded. Data instances from various attacks were significantly out of proportion to normal data. Parameters such as true positive rate (TPR) and false positive rate (FPR) were utilized to estimate the effectiveness of the approaches and comparative analysis. A partial view of the results of the six algorithms describing the comparative analysis of normal true positive rate, attack true positive rate, normal false positive rate, and attack false positive rate for different sizes of attribute sets of the KDDCUP'99 dataset [52] is presented in Table 2 and Figures 2–7, respectively.

The bar diagram of Figure 2 represents the percentage of normal true positive rate of six different algorithms, namely, Wang et al.'s algorithm [35], Cuijuan et al.'s algorithm [17], deep-RBF network [16], Bayes network [15], decision tree [14], and IFRSCAD for different attribute sizes, say 10, 20, and 41 of the dataset KDDCUP'99 [52]. Here, each colored bar represents one algorithm's percentage of normal true positive rates.

The bar diagram of Figure 3 represents the percentages of attack true positive rate of six aforesaid algorithms (Wang et al.'s algorithm [35], Cuijuan et al.'s algorithm [17], deep-RBF network [16], Bayes network [15], decision tree [14], and IFRSCAD) for different attribute sizes of the dataset KDDCUP'99 [52].

The bar diagram of Figure 4 represents the percentages of normal false positive rates of aforesaid algorithms for different attribute sizes of the dataset KDDCUP'99 [52].

Again the bar diagram of Figure 5 represents the percentages of the attack false positive rates of the aforesaid six algorithms for different attribute sizes of the dataset KDDCUP'99 [52].

Again, the bar diagram of Figure 6 represents the percentages of the average true positive rate of the aforesaid six algorithms for different attribute sizes of KDDCUP'99 [52].

**Table 2.** Normal vs. attack TPR/FPR using KDDCUP'99 [52].

| Algorithm | No. of Attributes | Normal TPR | Attack TPR | Normal FPR | Attack FPR | Avg. TPR | Avg. FPR |
|---|---|---|---|---|---|---|---|
| IFRSCAD | 41 | 0.9699 | 0.9629 | 0.03010 | 0.03710 | 0.9664 | 0.03360 |
| | 20 | 0.97999 | 0.9789 | 0.02010 | 0.02110 | 0.974445 | 0.02060 |
| | 10 | 0.98342 | 0.9804 | 0.01658 | 0.01960 | 0.98191 | 0.01809 |
| Wang et al. [35] | 41 | 0.9625 | 0.9325 | 0.0302 | 0.0675 | 0.9475 | 0.04885 |
| | 20 | 0.9745 | 0.9415 | 0.0312 | 0.0585 | 0.9580 | 0.04485 |
| | 10 | 0.9821 | 0.9621 | 0.0212 | 0.0379 | 0.9721 | 0.02955 |
| Cuijuan et al. [17] | 41 | 0.9325 | 0.8925 | 0.0580 | 0.1075 | 0.9175 | 0.08275 |
| | 20 | 0.9445 | 0.9245 | 0.0540 | 0.0755 | 0.9345 | 0.06475 |
| | 10 | 0.9775 | 0.9575 | 0.0320 | 0.0425 | 0.9675 | 0.03725 |
| Deep-RBF network | 41 | 0.9025 | 0.8525 | 0.0975 | 0.1475 | 0.8775 | 0.12250 |
| | 20 | 0.9212 | 0.8812 | 0.0788 | 0.1188 | 0.9012 | 0.09880 |
| | 10 | 0.9425 | 0.9023 | 0.0575 | 0.0975 | 0.9225 | 0.07750 |
| Bayes network | 41 | 0.9313 | 0.8349 | 0.0687 | 0.1651 | 0.8831 | 0.11690 |
| | 20 | 0.9429 | 0.8720 | 0.0571 | 0.1328 | 0.9075 | 0.09255 |
| | 10 | 0.9587 | 0.9087 | 0.0413 | 0.0913 | 0.9337 | 0.05215 |
| Decision tree | 41 | 0.6649 | 0.6223 | 0.3351 | 0.3771 | 0.6436 | 0.35610 |
| | 20 | 0.6829 | 0.6520 | 0.3171 | 0.3480 | 0.6779 | 0.33255 |
| | 10 | 0.7131 | 0.6744 | 0.2969 | 0.3256 | 0.69375 | 0.31125 |

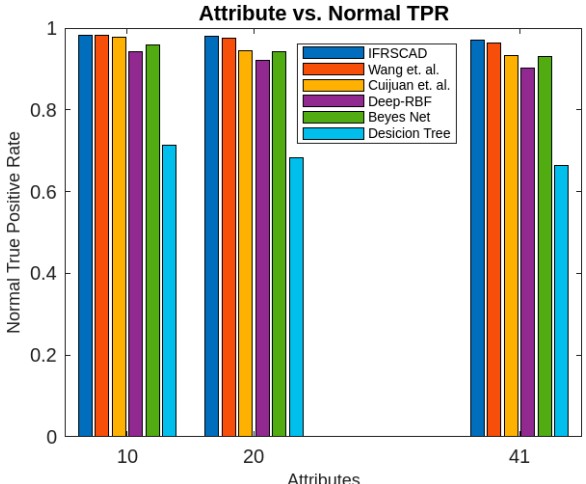

**Figure 2.** Comparative analysis of normal true positive rates of different algorithms with KDDCUP'99.

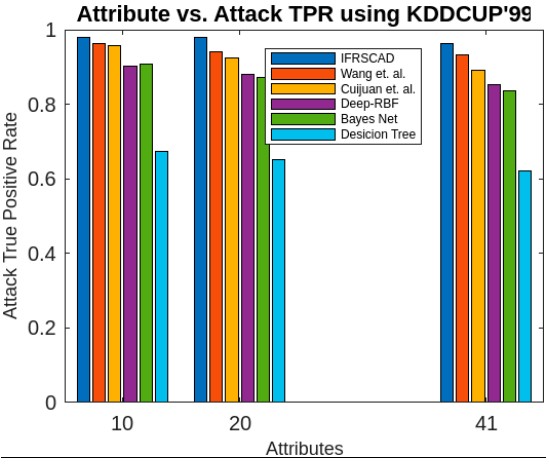

**Figure 3.** Comparative analysis of attack true positive rates of different algorithms with KDDCUP'99.

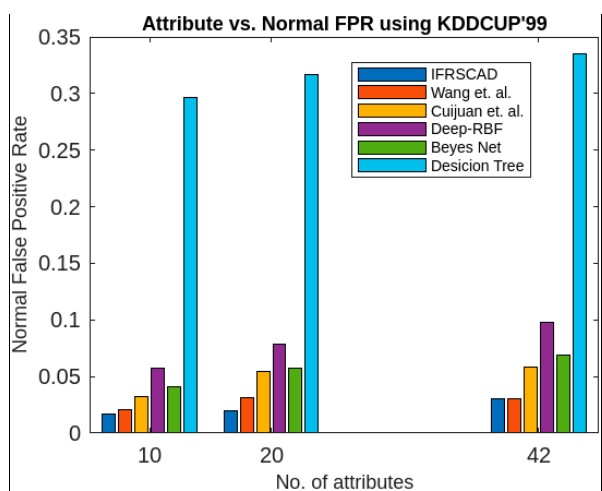

**Figure 4.** Comparative analysis of normal false positive rates of different algorithms with KDDCUP'99.

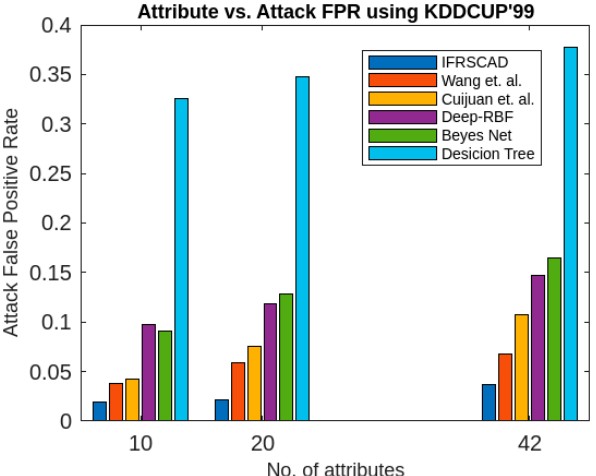

**Figure 5.** Comparative analysis of attack false positive rates of different algorithms with KDDCUP'99.

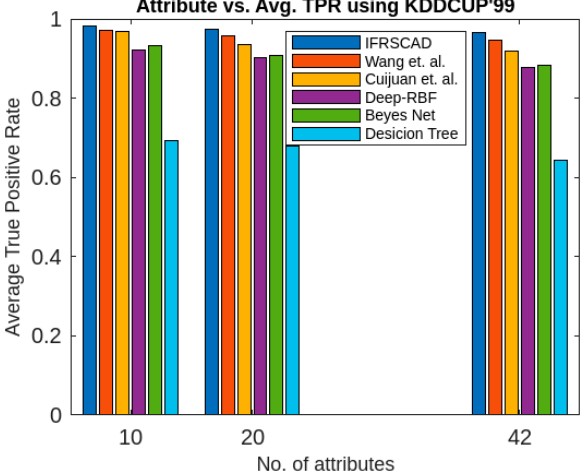

**Figure 6.** Comparative analysis of average true positive rates of different algorithms with KDDCUP'99.

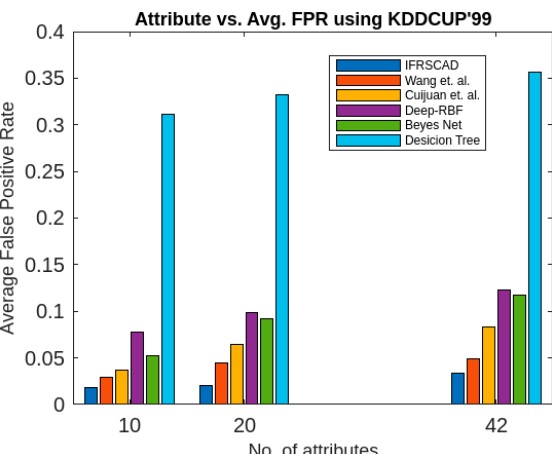

**Figure 7.** Comparative analysis of average false positive rates of different algorithms with KDDCUP'99.

Again, the bar diagram of Figure 7 represents the percentage of average false positive rate of the aforesaid six algorithms for different attribute sizes of the dataset KDDCUP'99 [52].

Similarly, a partial view of the results of the six algorithms describing the comparative analysis of normal true positive rate, attack true positive rate, normal false positive rate, and attack false positive rate for different sizes of attribute set of the Kitsune dataset [53] is presented in Table 3 and Figures 8–13, respectively.

**Table 3.** Normal vs. attack TPR/FPR using Kitsune [53].

| Algorithm | No. of Attributes | Normal TPR | Attack TPR | Normal FPR | Attack FPR | Avg. TPR | Avg. FPR |
|---|---|---|---|---|---|---|---|
| IFRSCAD | 115 | 0.91989 | 0.91289 | 0.08011 | 0.08711 | 0.91639 | 0.08361 |
| | 100 | 0.92766 | 0.92066 | 0.07234 | 0.07934 | 0.92416 | 0.07584 |
| | 50 | 0.9679 | 0.95116 | 0.04110 | 0.04884 | 0.95453 | 0.04497 |
| | 25 | 0.96999 | 0.96678 | 0.03010 | 0.03322 | 0.968385 | 0.03166 |
| | 10 | 0.98342 | 0.9804 | 0.01658 | 0.0196 | 0.98191 | 0.01809 |
| Wang et al. [35] | 115 | 0.9044 | 0.8933 | 0.0956 | 0.1067 | 0.89885 | 0.10115 |
| | 100 | 0.9277 | 0.9189 | 0.0723 | 0.0811 | 0.9233 | 0.1534 |
| | 50 | 0.9625 | 0.9425 | 0.0375 | 0.0575 | 0.9525 | 0.0475 |
| | 25 | 0.9745 | 0.9545 | 0.0255 | 0.0455 | 0.9645 | 0.0355 |
| | 10 | 0.9821 | 0.9621 | 0.0179 | 0.0379 | 0.9721 | 0.0279 |
| Cuijuan et al. [17] | 115 | 0.8232 | 0.8142 | 0.18232 | 0.1858 | 0.8187 | 0.06703 |
| | 100 | 0.8633 | 0.8621 | 0.1367 | 0.1379 | 0.8627 | 0.1373 |
| | 50 | 0.9025 | 0.9011 | 0.0975 | 0.0989 | 0.9018 | 0.0982 |
| | 25 | 0.9445 | 0.9345 | 0.0555 | 0.0645 | 0.9395 | 0.0600 |
| | 10 | 0.9595 | 0.9575 | 0.0405 | 0.0425 | 0.9585 | 0.0415 |
| Deep-RBF network | 115 | 0.8121 | 0.8056 | 0.1879 | 0.1944 | 0.0885 | 0.19115 |
| | 100 | 0.8411 | 0.8352 | 0.1589 | 0.1648 | 0.83815 | 0.16185 |
| | 50 | 0.9025 | 0.8933 | 0.0975 | 0.1067 | 0.8979 | 0.1021 |
| | 25 | 0.9212 | 0.9102 | 0.0788 | 0.0898 | 0.9157 | 0.0843 |
| | 10 | 0.9425 | 0.9311 | 0.0575 | 0.0689 | 0.9368 | 0.07750 |
| Bayes network | 115 | 0.8055 | 0.7953 | 0.1945 | 0.2047 | 0.8004 | 0.1996 |
| | 100 | 0.8432 | 0.8342 | 0.1568 | 0.1658 | 0.8387 | 0.1613 |
| | 50 | 0.9313 | 0.9349 | 0.0687 | 0.0651 | 0.9331 | 0.0669 |
| | 25 | 0.9429 | 0.9420 | 0.0571 | 0.0580 | 0.94245 | 0.05755 |
| | 10 | 0.9587 | 0.9480 | 0.0413 | 0.0520 | 0.95335 | 0.04665 |
| Decision tree | 115 | 0.5012 | 0.4934 | 0.4988 | 0.5056 | 0.4973 | 0.5027 |
| | 100 | 0.5434 | 0.5345 | 0.4566 | 0.4655 | 0.53895 | 0.46105 |
| | 50 | 0.6449 | 0.6323 | 0.3551 | 0.3677 | 0.6386 | 0.3614 |
| | 25 | 0.6729 | 0.6629 | 0.3271 | 0.3371 | 0.6679 | 0.3321 |
| | 10 | 0.7131 | 0.6744 | 0.2869 | 0.3256 | 0.69375 | 0.30625 |

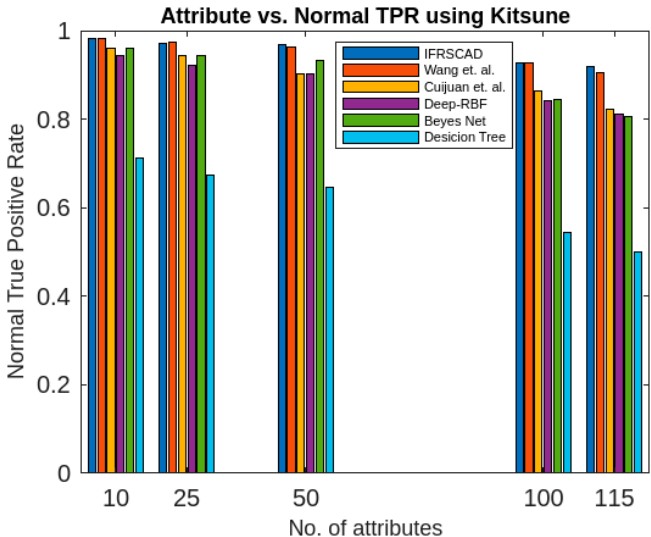

**Figure 8.** Comparative analysis of normal true positive rates of different algorithms with Kitsune [53].

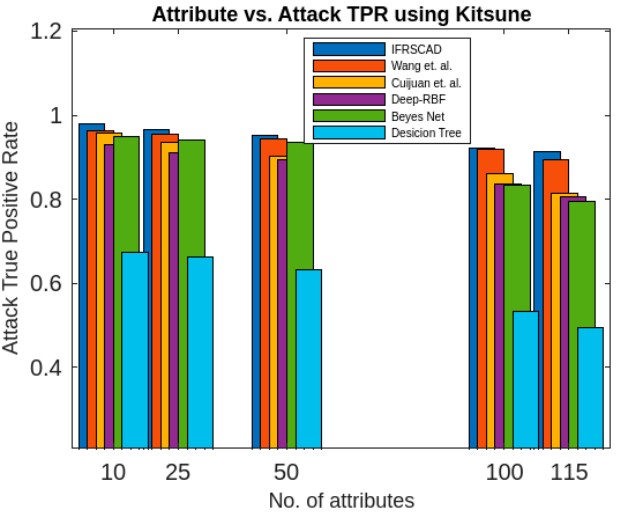

**Figure 9.** Comparative analysis of attack true positive rates of different algorithms using Kitsune.

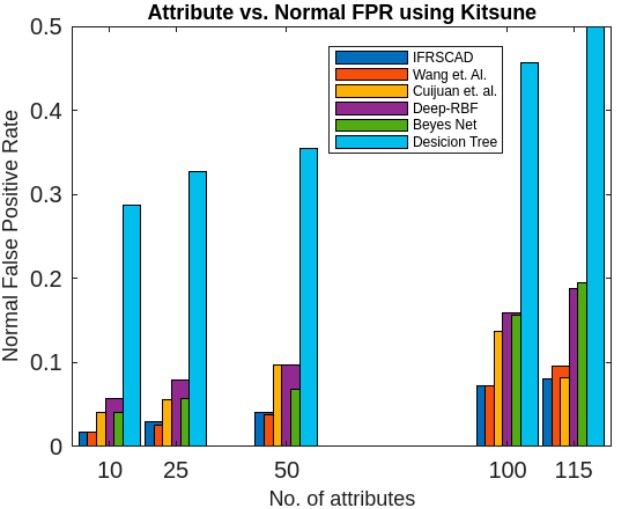

**Figure 10.** Comparing analysis of normal false positive rates of different algorithms with Kitsune.

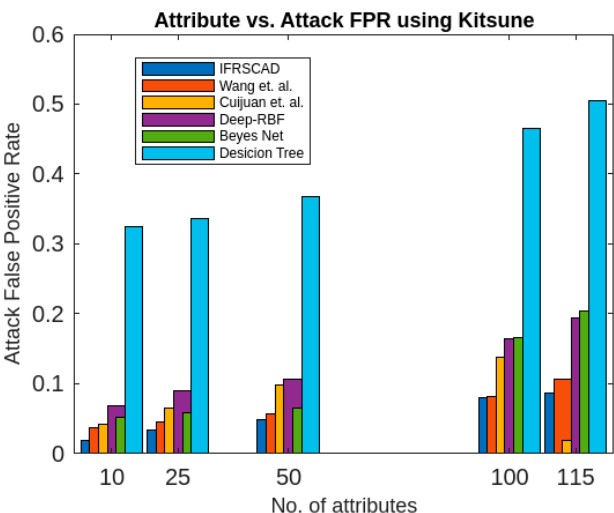

**Figure 11.** Comparing analysis of attack false positive rates of different algorithms with Kitsune.

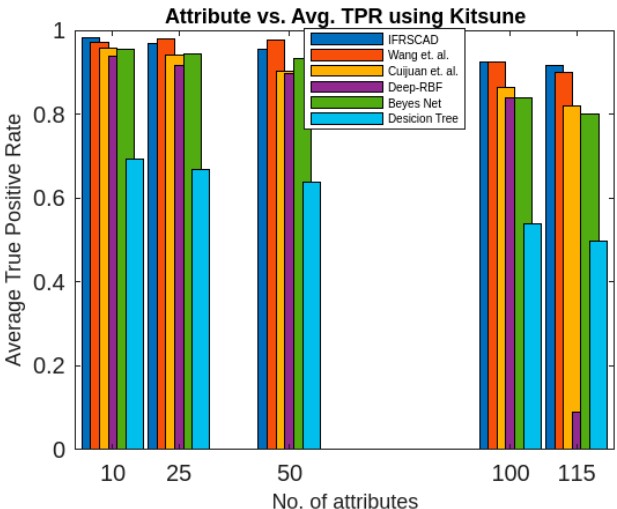

**Figure 12.** Comparing analysis of average true positive rates of different algorithms with Kitsune.

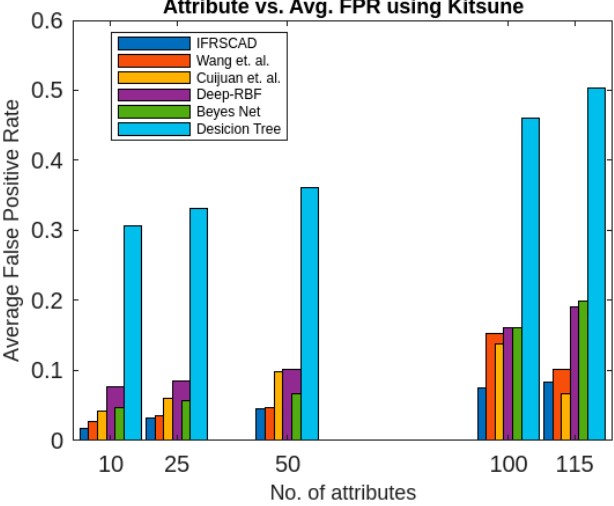

**Figure 13.** Comparing analysis of average false positive rates of different algorithms with Kitsune.

The bar diagram of Figure 8 represents the percentages of normal true positive rate of the aforesaid six algorithms (Wang et al.'s algorithm [35], Cuijuan et al.'s algorithm [17], deep-RBF network [16], Bayes network [15], decision tree [14], and IFRSCAD) for different attribute sizes, say 10, 25, 50, 100, and 41 of the dataset Kitsune [53]. Similarly, each colored bar represents one algorithm's percentage of normal true positive rates.

The bar diagram of Figure 9 represents the percentages of the attack true positive rate of six aforesaid algorithms (Wang et al.'s algorithm [35], Cuijuan et al.'s algorithm [17], deep-RBF network [16], Bayes network [15], decision tree [14], and IFRSCAD) for different attribute sizes of the dataset Kinsune [53].

The bar diagram of Figure 10 represents the percentages of normal false positive rates of the six aforesaid algorithms (Wang et al.'s algorithm [35], Cuijuan et al.'s algorithm [17], deep-RBF network [16], Bayes network [15], decision tree [14], and IFRSCAD) for different attribute sizes of the dataset Kitsune [53].

The bar diagram of Figure 11 represents the percentages of attack false positive rates of the six aforesaid algorithms (Wang et al.'s algorithm [35], Cuijuan et al.'s algorithm [17], deep-RBF network [16], Bayes network [15], decision tree [14], and IFRSCAD) for different attribute sizes of the Kitsune dataset [53].

The bar diagram of Figure 12 represents the percentages of average true positive rates of the aforesaid six algorithms (Wang et al.'s algorithm [35], Cuijuan et al.'s algorithm [17], deep-RBF network [16], Bayes network [15], decision tree [14], and IFRSCAD) for different attribute sizes of the dataset Kitsune [53].

Again, the bar diagram of Figure 13 represents the percentages of average true positive rates of the aforesaid six algorithms (Wang et al.'s algorithm [35], Cuijuan et al.'s algorithm [17], deep-RBF network [16], Bayes network [15], decision tree [14], and IFRSCAD) for different attribute sizes of the dataset Kitsune [53].

The following observations can be drawn from the above tables and bar diagrams.

- The decision-tree-based algorithm [14] has the poorest detection rate. It has 71.31–66.49% of normal TPR, 67.44–62.23% of attack TPR, 29.69–33.51% of normal FPR, and 32.56–37.71% of attack FPR for ascending order of attribute sizes (from 10–41) of the dataset KDDCUP'99 [52]. Similarly, it has 71.31–50.12% of normal TPR, 67.44–49.34% of attack TPR, 28.69–49.88% of normal FPR, and 32.56–50.56% of attack FPR for ascending order of attribute sizes (from 10–115) of the dataset Kitsune [53]. It shows that the algorithm has the poorest performances, which decreases with the increase in dimension size of the dataset.

- The deep-RBF-network-based algorithm [16] is better than the decision-tree-based algorithm [14] and it has 94.25–90.25% of normal TPR, 90.23–85.25% of attack TPR, 5.75–9.75% of normal FPR, and 9.75–14.75% of attack FPR for ascending order of attribute sizes (from 10–41) of the dataset KDDCUP'99 [52]. Similarly, it has 94.25–81.21% of normal TPR, 93.11–80.56% of attack TPR, 5.75–18.79% of normal FPR, and 6.89–19.44% of attack FPR for ascending order of attribute sizes (from 10–115) of the dataset Kitsune [53].

- The Bayes-network-based algorithm [15] is better than the decision-tree-based algorithm [14] and the deep-RBF-network-based algorithm [16] in terms of detection rates. It has 95.87–93.13% of normal TPR, 90.87–83.49% of attack TPR, 4.13–6.87% of normal FPR, and 9.136–16.51% of attack FPR for ascending order of attribute sizes (from 10–41) of the dataset KDDCUP'99 [52]. Similarly, it has 95.87–80.55% of normal TPR, 94.8–79.53% of attack TPR, 4.13–19.45% of normal FPR, and 5.20–20.47% of attack FPR for ascending order of attribute sizes (from 10–115) of the dataset Kitsune [53]. Although the algorithm is quite efficient, its performance decreases with the increase in the dimension of the datasets.

- Cuijuan et al.'s algorithm [17] is better than all the previous three algorithms as far as detection rate is concerned. It has 97.75–93.25% of normal TPR, 95.25–89.25% of attack TPR, 3.20–5.80% of normal FPR, and 4.25–10.75% of attack FPR for ascending order of attribute sizes (from 10–41) of the dataset KDDCUP'99 [52]. Similarly, it has

95.95–82.32% of normal TPR, 95.75–81.42% of attack TPR, 4.05–18.232% of normal FPR, and 4.25–18.58% of attack FPR for ascending order of attribute sizes (from 10–115) of the dataset Kitsune [53]. Its performance also decreases proportionately with the increase in the dimension of the datasets.

- Wang et al.'s algorithm [35] is the most efficient in comparison with all the aforesaid algorithms. It has 98.21–96.25% of normal TPR, 96.21–93.25% of attack TPR, 2.12–3.02% of normal FPR, and 3.79–6.75% of attack FPR for ascending order of attribute sizes (from 10–42) of the dataset KDDCUP'99 [52]. Similarly, it has 98.21–90.44% of normal TPR, 96.21–89.33% of attack TPR, 1.79–9.56% of normal FPR, and 3.79–10.67% of attack FPR for ascending order of attribute sizes (from 10–115) of the dataset Kitsune [53]. Its performance also decreases proportionately with the increase in the dimension of the datasets.

- The proposed algorithm (IFRSCAD) has 98.342–96.99% of normal TPR, 98.04–96.29% of attack TPR, 1.658–3.01% of normal FPR, and 1.96–3.71% of attack FPR for ascending order of attribute sizes (from 10–42) of the dataset KDDCUP'99 [52]. Similarly, it has 98.342–91.989% of normal TPR, 98.04–91.289% of attack TPR, 1.658–8.011% of normal FPR, and 1.96–8.711% of attack FPR for ascending order of attribute sizes (from 10–115) of the dataset Kitsune [53]. Its performance also decreases proportionately with the increase in the dimension of datasets. It is clear from the data that the proposed algorithm has more TPR and less FPR. The difference between normal TPR and attack TPR and normal FPR and attack FPR is also less in comparison with other methods. The performance decrement is less with the increase in dimensions. Obviously, the IFRSCAD has more average TPR and less average FPR than others.

- In addition, the execution time of the IFRSCAD depends upon two factors, namely, dimension and size of the datasets. It was found that if the dimension is kept constant, the algorithm has quadratic execution time, whereas if the data size is kept constant, it runs in linear time. Therefore, the proposed algorithm's time complexity is more dependent on the data size than the number of attributes. The time-complexity graphs for constant dimension and constant data size are given, respectively, in Figures 14 and 15.

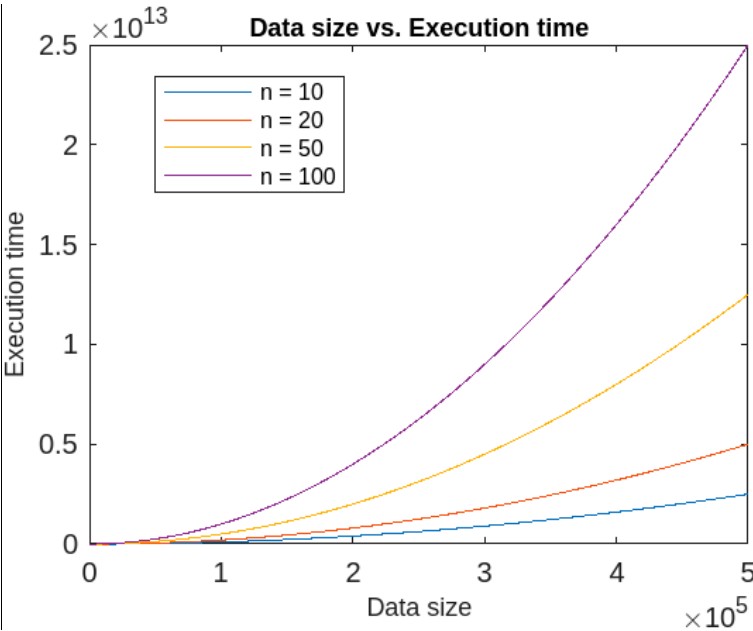

**Figure 14.** Execution time of IFRSCAD for different dimensions (*n*).

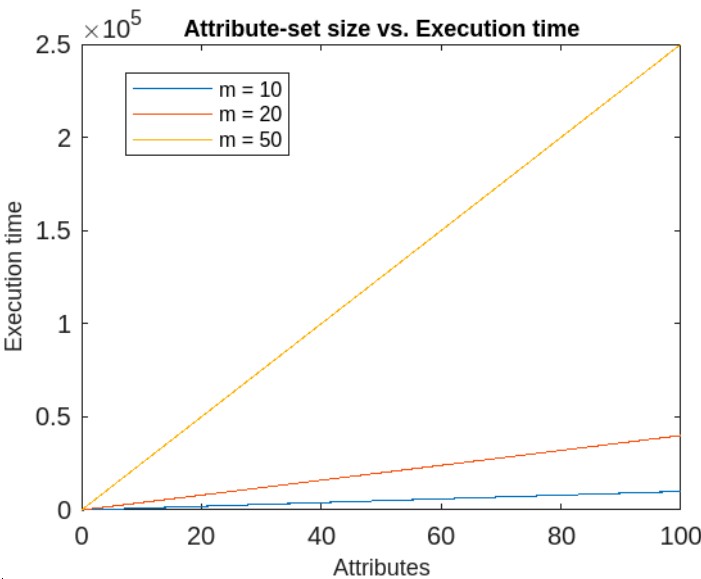

**Figure 15.** Execution time of IFRSCAD for different data sizes (*m*).

Furthermore, the IFRSCAD's time-complexity is also analyzed against that of MCAD [30]. If the dimension of the dataset is assumed to be constant, the MCAD [30] runs in cubic time and IFRSCAD runs in quadratic time. Thus for large data size, the IFRSCAD outperforms MCAD [30]. The comparative analysis is presented graphically in Figure 16 below.

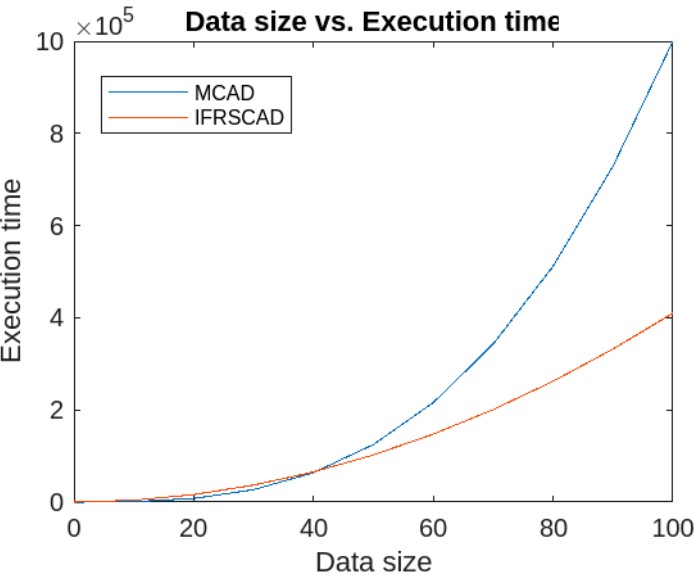

**Figure 16.** Comparative analysis of execution time of MCAD [30] and IFRSCAD.

## 7. Conclusions, Limitations, and Lines for Future Work

### 7.1. Conclusions

In this article, a hybrid algorithm consisting of both rough set and fuzzy set theoretic approaches is presented for the detection of anomaly. The algorithm is a classification-based algorithm which uses rough set and intuitionistic fuzzy set to deal with uncertainty in the dataset. The obtained rules can be expressed using intuitionistic fuzzy sets. The algorithm generates certain rules from lower approximation space, possible rules from upper approximation space, and boundary rules from boundary regions. Each attribute contributing in a rule can be expressed in terms of its membership value and non-membership value.

In addition, an attribute can contribute in both the certain rules as well as the possible rules. Therefore, each rule obtained by the algorithm is expressed using intuitionistic fuzzy set. The algorithm is named IFRSCAD. The proposed algorithm's performance is demonstrated by experimental analysis, and using the datasets KDDCUP'99 [52] and Kitsune [53], the algorithm extract anomalies with the accuracy of 96.99% and 91.989%, respectively. The comparative analysis shows that the proposed algorithm outperforms a couple of well-known classification-based algorithms.

Finally, the proposed algorithm's time-complexity is found to be less dependent on dimension of the dataset and, rather, more on the size of the datasets. However, the detection rate depends more on dimensions, as evident from the obtained results. The proposed algorithm's time-complexity is compared with a clustering-based algorithm MCAD [30], and under the assumption of constant dimension, the algorithm is found to be more efficient than MCAD [30].

### 7.2. Limitations and Lines for Future Work

Though the proposed algorithm performs very well, it has some limitations. Firstly, although the run time of the proposed algorithm is less dependent on dimension of the dataset, it detection rate decreases proportionately with the increase in dimension. Secondly, the algorithm lacks efficacy in dealing with continuous data, as rough set cannot handle continuous data, and finding the correlation coefficient of continuous data would be difficult. Finally, the algorithm in its current form is inefficient to deal with real-time data.

Future works can be possible along the following lines:

- An effective method can be designed for anomaly detection in high-dimensional data.
- An effective method can be designed for anomaly detection from datasets with continuous attributes.
- An effective method can be designed for real-time anomaly from heterogeneous data.

**Author Contributions:** Conceptualization, F.A.M.; Methodology, F.A.M.; Software, F.A.M., M.S.; Validation, F.A.M., M.S.; Formal Analysis, F.A.M.; Investigation, F.A.M., M.S.; Resource, F.A.M., M.S.; Data Curation, F.A.M., M.S.; Writing—original draft preparation, F.A.M., M.S.; writing—review and editing, F.A.M., M.S.; visualization, F.A.M.; supervision, F.A.M.; project administration, M.S., F.A.M.; funding acquisition, M.S. All authors have read and agreed to the published version of the manuscript.

**Funding:** This research received no external funding.

**Informed Consent Statement:** Not applicable.

**Data Availability Statement:** The data, code, and other materials can be made available on request.

**Conflicts of Interest:** There are no conflict of interest or competing interests among the authors. All the authors have agreed to publish the paper in this journal.

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
