# Peer review of "An Intuitionistic Fuzzy-Rough Set-Based Classification for Anomaly Detection"

_applsci, doi:10.3390/app13095578_

Round 1
Reviewer 1 Report
I have uploaded the comments for authors with an attachment.

Author Response
- This article is missing a lot of punctuation, for example, line 196, line 201, line 202, the symbol is missing after the sentence. Please correct many similar mistakes later.
Response1. The authors have carefully gone through the manuscript and corrected all the mistakes mentioned by honorable reviewer.
- In p1, line 15, “behaviour” should be revised to “behaviors”.
Response2. The authors have corrected it.
- In p2, line 46, “approach” should be revised to “approaches”, line 60, “authored” should be revised to “authors”
Response3. The authors have corrected it.
- In p3, line 117, “the Section 2” should be revised to “Section 2”.
Response4. The authors have corrected it.
- In p5, line 181, “KDD Cup 99” should be revised to “KDD CUP 99”.
Response5. The authors have corrected it.
- In p6, line 207, “a-cut” should be revised to “a-cut”, line 229, B1(X)) should be revised to B1(X).
Response6. The authors have corrected it.
- In p7, line 241, Section-3 should be revised to Section 3.
Response7. The authors have corrected it.
- In p8, line 289, “rules” should be revised to “rule”.
Response8. The authors have corrected it.
- The reference format should be consistent.
Response9. The authors made the reference consistent as advised by the honourable reviewer.
Reviewer 2 Report
The idea of proposes a hybrid approach based on rough set theory and intuitionistic fuzzy set theory is proposed for the detection of anomaly is interesting and useful. However, I have some comments that can help the authors improve the added value of the analysis in light of previous studies. I hope that my below comments allow the authors to improve their paper further.
-
Introduction: This section should be justified by recent studies that make it possible to demonstrate the knowledge gap. It has 43 references and very few from recent years. Also, please indicate the importance of using intuitionistic fuzzy set and rough sets for having a better a solution of problem stabled.
-
It is unclear the paper’s objective. Please rewrite the objective, this looks like a description of a process.
-
After reading the Abstract and Introduction section, it is unclear what the paper's main purpose is. Besides, what is the knowledge gap the paper intends to close? While presenting contribution, the authors should refer to previous studies and how their paper is different and add to the existing literature.
-
Have the authors named the proposed model as IFRSCAD?
-
Related works. Please compare, in a deeper way, the paper’s contribution with earlier studies. Thus, it also needs to be adjusted with studies published in recent years. Such an extension of the related literature will help you to connect your paper with the existing literature and add its ability to collect citations from the rising literature on anomaly detection. I suggest to use Scopus, WoS, and VosViewer.
-
Subsection B. Experimental results and analysis. Each figure must have a complete explication after each one is presented. Please make a table to compare the results.
-
The Conclusion section requires to be rewrite. Please include the main results in quantitative terms. Please don't use only bullets.
The authors should carefully review the entire text of the article and pay more attention to explaining the relate works, results, discussions, and conclusions. For example, the conclusions section is very weak and do not reflect the scope of the work, This does not signifi
Author Response
1. Introduction: This section should be justified by recent studies that make it possible to demonstrate the knowledge gap. It has 43 references and very few from recent years. Also, please indicate the importance of using intuitionistic fuzzy set and rough sets for having a better a solution of problem stabled.
Response1. The authors have revised the Introduction including more recent studies and indicating the importance of using intuitionistic fuzzy set and rough sets for having a better a solution of problem stabled as advised by the honorable reviewer.
2. It is unclear the paper’s objective. Please rewrite the objective, this looks like a description of a process.
Response2. The paper’s objective has been rewritten as suggested by the honorable reviewer.
3. After reading the Abstract and Introduction section, it is unclear what the paper's main purpose is. Besides, what is the knowledge gap the paper intends to close? While presenting contribution, the authors should refer to previous studies and how their paper is different and add to the existing literature.
Response3. The abstract and Introduction is revised to highlight the main purpose and knowledge gap by referring the previous studies and contribution of proposed work against others.
4. Have the authors named the proposed model as IFRSCAD?
Response4. Yes.
5. Related works. Please compare, in a deeper way, the paper’s contribution with earlier studies. Thus, it also needs to be adjusted with studies published in recent years. Such an extension of the related literature will help you to connect your paper with the existing literature and add its ability to collect citations from the rising literature on anomaly detection. I suggest to use Scopus, WoS, and VosViewer.
Response5. The related work section is revised by adding some very recent works.
6. Subsection B. Experimental results and analysis. Each figure must have a complete explication after each one is presented. Please make a table to compare the results.
Response6. The description of each figure is given after its presentation. Two tables Table 2 and Table 3 are included for comparative analysis.
7. The Conclusion section requires to be rewrite. Please include the main results in quantitative terms. Please don't use only bullets.
Response7. The conclusion section is rewritten including main results. Bullets are removed.
8. The authors should carefully review the entire text of the article and pay more attention to explaining the relate works, results, discussions, and conclusions. For example, the conclusions section is very weak and do not reflect the scope of the work, This does not signify
Response8. The authors have carefully reviewed the entire text done necessary explanation as suggested by the honorable reviewer.
Reviewer 3 Report
The model described in reference 28 should be included in the comparative analysis, since it has been applied by the authors to the same datasets. Moreover, the competing rivals hosted in 28 could also be included.
Additionally, the authors have dealt with the same problem in references 8,9,12 and 20. Some of the results presented in these papers could also be included.
Lines 19 and 456: "both rough and ..." -> "of both rough and..."
Author Response
1. Comments and Suggestions for Authors: The model described in reference 28 should be included in the comparative analysis, since it has been applied by the authors to the same datasets. Moreover, the competing rivals hosted in 28 could also be included. Additionally, the authors have dealt with the same problem in references 8,9,12 and 20. Some of the results presented in these papers could also be included.
Response1. The authors have included the comparative time-complexity analysis of the proposed method with reference 28. Since the references 8, 9, 20, and 28 is a clustering-based approaches and proposed approach being classification-based approach authors have ignored other comparative analysis. However, the authors have included five other similar approaches for comparative analysis. Also, two datasets Kitsune and KDDCUP’99 have been used for experiments but in other cases only KDDCUP’99 is used.
2. Comments on the Quality of English Language: Lines 19 and 456: "both rough and ..." -> "of both rough and...".
Response2. The authors have checked the text carefully and corrected the mistakes as suggested by the honorable reviewer.
Reviewer 4 Report
This manuscript investigates the issue of anomaly detection. A hybrid method integrating rough set theory and intuitionistic fuzzy set theory is proposed. Some comments are given as follows.
1. The main novelty and contribution of this manuscript are not clear.
2. In the Abstract, what is the meaning of “softness properties” in line 19?
3. In line 40, anomaly detection-based and signature recognition-based techniques are mentioned. However, only former is explained while the latter is not.
4. For the anomaly detection issue, more relative literature such as 10.1109/JIOT.2019.2958185 and 10.1080/00207721.2022.2143735 should be reviewed.
5. The advantage of the proposed algorithm is not clear. And the arrows in Figure 1 is very chaotic.
6. Section 5 gives the complexity analysis. However, there lacks a comparison analysis to illustrate that the proposed algorithm is quite efficient.
Author Response
1. The main novelty and contribution of this manuscript are not clear.
Response1. Abstract and Introduction are revised to make the novelty and contribution clear.
2. In the Abstract, what is the meaning of “softness properties” in line 19?
Response1. Abstract is rewritten. The soft property is used to indicate the uncertainty, imprecision, and vagueness associated with fuzzy and rough set.
3. In line 40, anomaly detection-based and signature recognition-based techniques are mentioned. However, only former is explained while the latter is not.
Response3. The signature recognition-based technique is explained in the revision.
4. For the anomaly detection issue, more relative literature such as 10.1109/JIOT.2019.2958185 and 10.1080/00207721.2022.2143735 should be reviewed.
Response4. The above two along with more recent articles are included in literature survey.
5. The advantage of the proposed algorithm is not clear. And the arrows in Figure 1 is very chaotic.
Response5. The authors have made the advantages clear. The Figure 1 is corrected.
6. Section 5 gives the complexity analysis. However, there lacks a comparison analysis to illustrate that the proposed algorithm is quite efficient.
Response6. The complexity of the algorithm is compared with a new algorithm MCAD.
Round 2
Reviewer 2 Report
I am grateful to the authors for the changes made to the article. The corrections significantly improved the understanding of the study's problem, methods and results. I believe that in this form the article can be published.
Reviewer 3 Report
The authors put a lot of effort to review their work.
Just a quick editing to correct some typos.